

# Prevalence of type 2 diabetes mellitus and hypertension in patient's visiting the conservative dentistry and endodontics department: a cross-sectional study in Surabaya City

Meet Manihar[1], Dian Agustin Wahjuningrum[2], Shreya Manihar[3], Ajinkya M. Pawar[1], Jatin Atram[1], Kulvinder Banga[1], Alexander Maniangat Luke[4,5] and Firas Elmsmari[4,5]

[1] Department of Conservative Dentistry and Endodontics, Nair Hospital Dental College, Mumbai, Maharashtra, India

[2] Department of Conservative Dentistry, Universitas Airlangga, Surabaya, Indonesia

[3] Internal Medicine, St James University Hospital, Leeds, United Kingdom

[4] Department of Clinical Sciences, College of Dentistry, Ajman University, Ajman, United Arab Emirates

[5] Center for Medical and Bio-Allied Health Sciences Research (CMBAHSR), Ajman University, Ajman, United Arab Emirates

Corresponding authors
Dian Agustin Wahjuningrum, dian-agustin-w@fkg.unair.ac.id
Ajinkya M. Pawar, ajinkya@drpawars.com

## ABSTRACT

**Background**. This descriptive cross-sectional study focuses on the prevalence of hypertension (HTN) and type 2 diabetes mellitus (T2DM) amongst patients who visited the Conservative Dentistry and Endodontics department. Recognizing these incidence statistics is critical for improving endodontic therapy delivery and assuring high-quality dental care with positive treatment outcomes.

**Methods**. In advance of getting dental care, all patients visiting the department were advised to get their blood sugar and blood pressure levels checked at random. Measurements were taken with digital equipment, and individuals with high levels were encouraged to seek medical advice before undergoing dental procedures. The obtained data was imported into Excel and analyzed with IBM SPSS software (version 21).

**Results**. The investigation had 1,100 participants (55.8% female and 44.2% male), with an average age of $44.58 \pm 12.77$ years. Of the individuals, 40.6% were referred for type 2 diabetes, 12.6% for hypertension, and 24.0% for both diseases. There was a significant correlation ($p < 0.05$) between referral status and gender. The average blood pressure and random blood sugar readings were 141.02 mmHg $\pm$ 56.28 mmHg (systolic), 79.83 mmHg $\pm$ 10.68 mmHg (diastolic), and 126.68 mg/dL $\pm$ 15.36 mg/dL, respectively. There was a substantial ($p < 0.05$) difference in mean systolic blood pressure between men and women. Furthermore, age was strongly connected with random blood sugar levels ($p < 0.05$) and systolic and diastolic blood pressure ($p < 0.05$). There were significant ($p < 0.05$) variations in mean blood pressure and blood sugar levels between referred and non-referred individuals.

**Conclusion**. Age had a relationship with higher random blood sugar levels, systolic blood pressure, and diastolic blood pressure. Dentists should consider patient age while planning treatment, as type 2 diabetes mellitus and hypertension require unique techniques to emphasize patient safety and produce excellent outcomes.

## INTRODUCTION

The increased incidence of chronic illnesses, notably type 2 diabetes mellitus (T2DM) and hypertension (HTN), pose a major challenge to global healthcare systems. Their ubiquitous incidence and serious repercussions for morbidity and mortality need a more in-depth investigation of their effects on different health aspects, including oral health. Type 2 diabetes mellitus is defined by its chronic metabolic nature and involves inappropriately elevated blood glucose levels; fasting glucose greater than 126 mg/dL, random blood glucose over 200 mg/dL, or hemoglobin A1C (HbA1c) exceeding 6.5% with or without antibodies to glutamic acid decarboxylase (GAD) and insulin (*Chi et al., 2021*). Hypertension is defined when systolic blood pressure is consistently ≥130 mm Hg and/or diastolic blood pressure is ≥80 mm Hg (*Flack & Adekola, 2020*).

An important topic of research with implications on treatment is the reciprocal interaction between type 2 diabetes mellitus (DM), hypertension (HTN), and dental health in the context of endodontic therapy. Endodontics, a profession dedicated to the effective diagnosis and treatment of pulp pathosis, requires careful consideration of these comorbidities as there exists a bidirectional relationship between these comorbidities and oral health. Type 2 diabetes mellitus has been associated with an increased vulnerability to periodontal infections, which can result in tooth loss.

Studies suggest that patients with type 2 diabetes mellitus and hypertension can have a greater incidence of oral health problems, which could affect how well endodontic therapy works (*Cintra et al., 2021*; *Niazi & Bakhsh, 2022*; *Segura-Egea et al., 2023*).

Chronic metabolic disease known as type 2 DM raises the incidence of endodontic pathosis and apical periodontitis (AP), among other oral health issues. According to studies, people with diabetes mellitus may require endodontic therapy more frequently and have systemic illnesses including hypertension, obesity, coronary artery disease, and hyperlipidemia more frequently (*Chauhan et al., 2023*). This correlation emphasizes how crucial it is to take patients' systemic health state into account while they are receiving endodontic therapy. Furthermore, the connection between dental health, HTN, and T2DM goes beyond the course of therapy. An increased incidence of endodontic failures and a propensity for non-healing resulting from endodontic operations have been associated with poorly managed type 2 diabetes mellitus (*León-López et al., 2023*). Furthermore, patients with type 2 DM may exhibit an increased prevalence of cardiovascular disorders, underscoring the importance of a thorough systemic assessment prior to beginning endodontic therapy (*De Araujo et al., 2024*). However, dental health can also affect how T2DM and HTN are managed. Research indicates that dental conditions may have systemic effects, influencing diabetes patients' glucose control and perhaps exacerbating the consequences of hypertension (*González-Moles & Ramos-García, 2021*).

Therefore, in order to provide patients seeking endodontic treatment with the best care possible, a comprehensive strategy that takes into account the reciprocal interaction

between systemic illnesses like DM and HTN, oral health status, and endodontic therapy is essential.

Similarly, HTN has been associated to poor periodontal health and suboptimal dental operation results (*Petrie, Guzik & Touyz, 2018*; *Khan et al., 2019*; *Han, Son & Kim, 2021*). Oral infections and inflammation resulting from untreated endodontic problems can exacerbate systemic conditions like T2DM and HTN, as the body's inflammatory response may contribute to insulin resistance and elevated blood pressure Individuals with uncontrolled T2DM often exhibit compromised immune function, impaired wound healing, and increased susceptibility to oral infections. Similarly, hypertension can lead to vascular changes, affecting oral tissues' blood supply and healing processes. Additionally, hypertension-induced vascular changes may affect tissue oxygenation and nutrient delivery, further hampering healing processes.

This association between T2DM and HTN needs a more in-depth investigation with different variables like age, gender, etc. of the patients for providing an effective treatment protocol. It is the utmost responsibility of the medical, dental and paramedical staff to thoroughly understand the complexity of these diseases in order to provide optimum patient care. Thus, comprehensive research efforts are required to unravel the underlying processes of these connections. This research will eventually inform the creation of individualized treatment options to reduce the impact of chronic diseases on dental health and general well-being.

Indonesia, among many developing nations, confronts the daunting task of controlling the increasing and onerous prevalence of T2DM and HTN. The country's health data provide a bothering envision: hypertension prevalence was 31.8% in 2013 and is expected to rise in the near future, while type 2 diabetes prevalence was 10.9% in 2018 and is expected to reach 16.7% by 2030 (*Carey & Whelton, 2018*; *Setiadi et al., 2022*; *Zainuddin et al., 2023*; *Soeatmadji et al., 2023*). This concerning trend is especially noticeable in Surabaya, a highly populated city in East Java, where chronic illnesses like T2DM and HTN pose a serious public health danger.

While the effects of type 2 diabetes and hypertension on oral health are well recognized, there is little information on their incidence among Indonesian patients seeking endodontic therapy. This knowledge gap is of considerably significant in the Department of Conservative Dentistry, where more complicated and difficult dental procedures such as tooth caries and pulpitis are treated. To improve oral health and general well-being of the patient, it is critical to address this gap and create individualized treatment programs that maximize benefits while minimizing risks in this vulnerable patient population.

By examining the prevalence of T2DM and HTN among patients seeking dental care at the Department of Conservative Dentistry in Surabaya, this cross-sectional study seeks to close this gap. In particular, we want to:

1. Find out how common T2DM and HTN are among patients in Surabaya who are receiving endodontic and conservative dental care.
2. Examine any connections that could exist between T2DM/HTN and certain dental procedures the department offers.

Keeping the study objective in mind and looking at the anticipated rise in the incidence of T2DM and HTN, the authors through this study put forward the hypothesis that there will be an elevated prevalence of T2DM and HTN among the patients visiting the department for conservative and endodontic therapy.

## MATERIALS & METHODS

### Ethical approval statement

This study was undertaken and executed in adherence with the principles of the Helsinki Declaration, guaranteeing the welfare, rights, and privacy of all participants. The Universitas Airlangga Faculty of Dental Medicine Health Research Ethical Clearance Commission reviewed and approved the research protocol, which received the reference number 1176/HRECC.FODM/X/2023. All participants supplied informed permission after being fully told about the study's objective, methods, potential risks, and benefits. Throughout the research procedure, participants' confidentiality and identities were scrupulously preserved.

### Study setting

This study was conducted among the patients visiting the Department of Conservative Dentistry and Endodontics. The location of the study was Department of Conservative Dentistry and Endodontics, Universitas Airlangga, Surabaya City, Indonesia. The data collection for the study was carried out for 6 months starting from 10th October, 2023 to 10th April, 2024.

### Sample size estimation

Using the formula [DEFF*Np(1-p)]/ [(d2/Z21- $\alpha$/2*(N-1)+p*(1-p)], the sample size was calculated. where the predicted frequency of 63.7%* at 99.9% confidence level with.1% precision(d) is the design effect (for cluster surveys-DEFF), which is 1. An estimated 1,001 people made up the entire sample. However, 1,100 subjects were included in the study.

## PARTICIPANTS

### Eligibility criteria

All the patients were advised to evaluate their blood sugar and blood pressure levels before proceeding with the dental treatment. Those who consented for the same and got their blood pressure and blood sugar levels evaluated were the subject of the current investigation. Patients from the city of Surabaya and surrounding areas make up our institute's diversified patient pool.

### Inclusion and exclusion criteria

Every patient visiting the department for conservative and endodontic treatment was advised to get their Blood sugar and Blood Pressure evaluated prior to the treatment. All those patients that consented and followed the advice were a part of the current research. All the patients who feared unnecessary complications like needle pricks, *etc.* and refrained from following the advice were excluded from the study.
## Demographic data

Every patient who undertook the evaluation was a resident of Surabaya City, Indonesia.

## Variables

The variables under consideration in the current descriptive study are blood pressure and random blood sugar.

    Patients were categorized into two groups:

1. Normal patients–Patients whose blood pressure and random blood sugar levels fall within the normal range (below 130/90 mmHg for blood pressure and below 126 mg/dL for random blood sugar) and who were considered suitable to proceed with dental treatment.
2. Referred patients–Patients with blood pressure levels greater than130/90 mm Hg (hypertensive) and random blood sugar levels greater than 126 gm/dl (diabetic) who were advised to seek medical interventions before proceeding with the dental treatment.

    Variables like pulpal pathosis, periapical conditions and other diseases were not considered as they were not the objective of the research.

## Diagnostic criteria

Patients having blood pressure levels greater than 130/90 mm of Hg were classified as hypertensive patients. Patients having random blood sugar levels greater than 126 mg/dl were classified as diabetic patients.

## Data measurement

### Recording and classifying patient's blood pressure levels

Patients were instructed to relax for 15 min before assessing the blood pressure. While assessing the blood pressure, patients were instructed to sit comfortably in an upright position while their blood pressure was measured using a digital sphygmomanometer (BPL Blood Pressure Monitor, Bengaluru, India), and the results were reported and categorised according to classification shown in (Table 1) (*Carey & Whelton, 2018*). If the blood pressure readings were less than 130/90 mm Hg, patients were informed to carry out their treatment. However, if the readings were higher than 130/90, the patients were instructed to unwind comfortably for 30 min before having their blood pressure checked again. The results were once again noted, and the average was taken into account. If the average readings still remained 130/90 mm Hg and above, the patients were advised to visit a nearby hospital for the appropriate control of blood pressure before returning for their dental treatment.

### Recording and classifying patient's plasma glucose levels

For recording Blood sugar levels, the patients were made to sit in comfortable position and the readings were recorded with the help of a glucometer (Glucospark glucometer, Sensa core Medical Instrumentation, Telangana, India). If the readings were less than 126 gm/dl, the patients were informed to continue with the treatment. If the reading were above 126 gm/dl, the patients were asked to visit a nearly hospital for the appropriate control of blood sugar before returning for their dental treatment.

**Table 1 American Heart Association classification of blood pressure.** Based on accurate measurements and average of $\geq 2$ readings on $\geq 2$ occasions.

| Category | Blood pressure measurements |
| --- | --- |
| Normal | $\leq$ 120/80 mm Hg |
| Elevated | 120–129/<80 mm Hg |
| Stage 1 hypertension | 130–139/80–89 mm Hg |
| Stage 2 hypertension | $\geq$ 140/90 mm Hg |

*Follow up*

Patients with elevated blood pressure and blood sugar levels were referred to the hospital for specialized treatment and management of these conditions. Upon assessment, their medication doses were modified to effectively manage these conditions. Through meticulous monitoring and adjustments, the medical team successfully achieved control over both elevated blood pressure and blood sugar levels. With the health of the patient's stabilized, they were advised to return for dental treatment, ensuring comprehensive care and optimal well-being. Following the referral for medical interventions, there was no subsequent follow up or data collected regarding the patient's return for dental treatment as it was not within the scope of the research objectives.

## Statistical analysis

The data was accurately entered from the departmental records in Microsoft Excel Version 13 and was sent for statistical analysis (IBM SPSS version 21). Frequency percentage was obtained for the categorical variable (gender and, referral) and for continuous variable (systolic blood pressure, diastolic blood pressure and random blood sugar level) mean SD was obtained. To compare systolic blood pressure, diastolic blood pressure and random blood sugar level between gender unpaired T test was applied. To compare between systolic blood pressure, diastolic blood pressure and random blood sugar level between referral of the patients ANOVA was applied. Pearson correlation was done to observe the correlation between age and systolic blood pressure, diastolic blood pressure and random blood sugar Level of the patients. All the tests were applied keeping confidence interval at 95% and ($p < 0.05$) was considered to be statistically significant.

## RESULTS

### Demographic data

For the current study, the minimum age of the participants was 17 years and the maximum age was 90 years. The mean age of the participants was $44.58 \pm 12.77$. Of the 1,100 people, 486 (44.2%) were men and 614 (55.6%) were women. Of these, 447 (40.6%) were referred to the hospital as type 2 diabetic patients, 264 (24.0%) as hypertension patients, and 250 (22.7%) as normal patients. Furthermore, the medical hospital received referrals for hypertension alone from 139 patients (12.6%). Regarding geography and referral proportions, there were significant differences in the distribution ($p = 0.00$) (Table 2).

**Table 2  Distribution of participants depending upon gender and referral.**

|  |  | Frequency (n) | Percentage (%) | *P* value |
|---|---|---|---|---|
| Gender | Male | 486 | 44.2 | 0.00 |
|  | Female | 614 | 55.8 |  |
| Referral | Normal Patients | 250 | 22.7 |  |
|  | Referred to Hospital for Diabetes | 447 | 40.6 |  |
|  | Referred to Hospital for Hypertension | 139 | 12.6 | 0.00 |
|  | Referred to Hospital for Diabetes and Hypertension | 264 | 24.0 |  |

**Table 3  Mean distribution of systolic diastolic and random blood sugar (gm/dl).**

|  | N | Minimum | Maximum | Mean | Std. Deviation |
|---|---|---|---|---|---|
| Systolic Blood pressure (in mmHg) | 1100 | 84.00 | 220.00 | 126.6818 | 15.36794 |
| Diastolic blood pressure (in mmHg) | 1100 | 48.00 | 134.00 | 79.8318 | 10.68687 |
| Random blood sugar (gm/dl) | 1100 | 48.00 | 560.00 | 141.0282 | 56.28163 |

## Distribution of systolic, diastolic and random blood sugar (gm/dl)

The recorded minimum and maximum systolic and diastolic pressure values were 84 mm Hg and 48 mm Hg, respectively, and 220 mm Hg and 134 mm Hg, respectively. The range of blood sugar values was 48 gm/dl to 560 mm Hg (Table 3).

## Comparison of systolic, diastolic blood pressure and random blood sugar levels

Table 4 juxtaposes random blood sugar levels and systolic and diastolic blood pressure according to the participant's gender. The results show a statistically significant difference between males and females in mean systolic blood pressure, with a difference of 3.39372 ($p = 0.000$). However, neither the mean diastolic blood pressure nor the random blood sugar levels showed a statistically significant relationship with gender.

## Correlation between age, systolic, diastolic blood pressure and random blood sugar levels

A positive correlation was found between age and systolic blood pressure, diastolic blood pressure and random blood sugar levels (Table 5). This correlation was statistically significant ($p = 0.000$). Figures 1A (systolic blood pressure and age) and 1B (diastolic blood pressure and age) represent the correlation.

**Table 4  Comparison of systolic and diastolic BP, random blood sugar levels.**

|  |  | Gender | N | Mean | Std. Deviation | Std. Error Mean | Mean Difference | T | *P* Value |
|---|---|---|---|---|---|---|---|---|---|
| Gender | Systolic blood pressure (mmHg) | Male | 486 | 128.5761 | 15.15900 | .68763 | 3.39372 | 3.658 | .000 |
|  |  | Female | 614 | 125.1824 | 15.37883 | .62064 |  |  |  |
|  | Diastolic blood pressure (mmHg) | Male | 486 | 80.3889 | 11.14640 | .50561 | .99801 | 1.539 | .124 |
|  |  | Female | 614 | 79.3909 | 10.29644 | .41553 |  |  |  |
|  | Random blood sugar (gm/dl) | Male | 486 | 143.1049 | 60.33360 | 2.73679 |  |  |  |
|  |  | Female | 614 | 139.3844 | 52.84702 | 2.13273 | 3.72057 | 1.089 | .276 |
|  |  | Non-MMR | 214 | 142.5888 | 54.49511 | 3.72521 |  |  |  |

**Table 5  Correlation of age with systolic diastolic and random blood sugar levels.**

|  |  | Systolic Blood pressure (mmHg) | Diastolic blood pressure (mmHg) | Random blood sugar (gm/dl) |
|---|---|---|---|---|
| Age (in years) | Pearson Correlation | .2325[**] | .099[**] | .212[**] |
|  | P Value | .000 | .001 | .000 |
|  | N | 1100 | 1100 | 1100 |

**Notes.**
[**]Positive correlation.

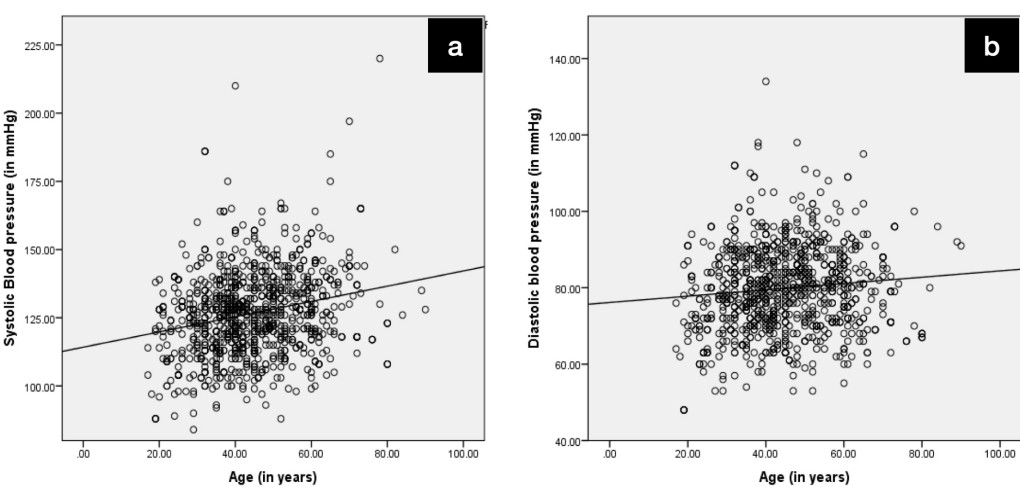

**Figure 1  Depicting positive correlation (A) between systolic blood pressure and age and (B) between diastolic blood pressure and age.**

## Comparison of systolic, diastolic, and random blood sugar levels with referral of the patients

Table 6 shows us that the comparison of systolic, diastolic and random blood sugar levels with referral of the patients depicted that there was statistically significant difference between mean systolic, diastolic and random blood sugar levels between the patients referred ($p = 0.000$).

**Table 6** Comparison of systolic, diastolic and random blood sugar levels with referral of the patients.

| | | N | Minimum | Maximum | Mean | SD | F, P Value |
|---|---|---|---|---|---|---|---|
| Systolic blood pressure (mmHg) | Normal Patient | 250 | 84.00 | 139.00 | 116.5600 | 10.43619 | |
| | Referred to Hospital for Diabetes | 447 | 93.00 | 167.00 | 119.0067 | 9.25462 | |
| | Referred to Hospital for Hypertension | 139 | 127.00 | 210.00 | 143.5755 | 13.38233 | 434.910 0.00 |
| | Referred to Hospital for both Diabetes and Hypertension | 264 | 105.00 | 220.00 | 140.3674 | 10.40279 | |
| Diastolic blood pressure (mmHg) | Normal Patient | 250 | 48.00 | 98.00 | 75.0200 | 9.50732 | |
| | Referred to Hospital for Diabetes | 447 | 53.00 | 110.00 | 76.3065 | 8.59724 | |
| | Referred to Hospital for Hypertension | 139 | 63.00 | 134.00 | 87.9353 | 11.66949 | 118.870, 0.00 |
| | Referred to Hospital for both Diabetes and Hypertension | 264 | 65.00 | 118.00 | 86.0909 | 8.81535 | |
| Random blood sugar (gm/dl) | Normal Patient | 250 | 48.00 | 130.00 | 103.1160 | 12.77825 | |
| | Referred to Hospital for Diabetes | 447 | 110.00 | 560.00 | 160.7785 | 63.53007 | |
| | Referred to Hospital for Hypertension | 139 | 52.00 | 173.00 | 102.9496 | 13.07355 | 121.536, 0.00 |
| | Referred to Hospital for both Diabetes and Hypertension | 264 | 110.00 | 375.00 | 163.5379 | 53.32573 | |

For a variety of causes, a statistically significant difference in Mean was found when a Pairwise Comparison of Systolic Blood Pressure with Patient Referral was carried out. Individuals with both hypertension and diabetes had higher systolic blood pressure ($p < 0.05$) (Table 7). With the exception of Normal Patients, those referred for Diabetes or Hypertension alone, and those referred for Both Hypertension and Diabetes ($p > 0.05$), the mean difference in diastolic blood pressure among patients referred to the hospital for various reasons was statistically significant ($p < 0.05$) (Table 8). A statistically significant difference ($p < 0.05$) was seen in the mean RBS between patients referred for hypertension, diabetes, and both diseases (excluding normal individuals) when random blood sugar levels were compared pairwise with patient referral (Table 9).

## DISCUSSION

The results of our investigations in Surabaya City illuminated an ominous reality. It provides a guiding light of the subtle hazards of chronic conditions like HTN and T2DM on effectiveness on the endodontic therapy. Our research unexpectedly revealed a concerning prevalence rate of 24.0% for HTN and 40.6% for type 2 DM in the studied population. Additionally, a positive correlation was found between age and systolic blood pressure, diastolic blood pressure and random blood sugar levels.

**Table 7  Pairwise comparison of systolic blood pressure with referral of the patients.**

|  |  | Normal patient | Referred to hospital for diabetes | Referred to hospital for hypertension | Referred to hospital for both diabetes and hypertension |
|---|---|---|---|---|---|
| Normal patient | Mean difference | – | −2.45[*] | −27.0[***] | −23.81[***] |
|  | p-value | – | 0.016 | <.001 | <.001 |
| Referred to hospital for diabetes | Mean difference |  | – | −24.6[***] | −21.36[***] |
|  | p-value |  | – | <.001 | <.001 |
| Referred to hospital for hypertension | Mean difference |  |  | – | 3.21[*] |
|  | p-value |  |  | – | 0.017 |

Notes.
[*] Non significant.
[**] Significant.
[***] Highly significant.

**Table 8  Pairwise comparison of diastolic blood pressure with referral of the patients.**

|  |  | Normal patient | Referred to hospital for diabetes | Referred to hospital for hypertension | Referred to hospital for both diabetes and hypertension |
|---|---|---|---|---|---|
| Normal patient | Mean difference | – | −1.29 | −12.9[***] | −11.07[***] |
|  | p-value | – | 0.297 | <.001 | <.001 |
| Referred to hospital for diabetes | Mean difference |  | – | −11.6[***] | −9.78[***] |
|  | p-value |  | – | <.001 | <.001 |
| Referred to hospital for hypertension | Mean difference |  |  | – | 1.84 |
|  | p-value |  |  | – | 0.231 |

Notes.
[*] Non significant.
[**] Significant.
[***] Highly significant.

Our research findings emphasize the significance of pre-assessing the prevalence and impact of T2DM and HTN within the realm of conservative dentistry and endodontics. This underscores the importance of evaluating these comorbidities prior to initiating the dental treatment.

Studies on T2DM and HTN, which reveal the intricate interactions between chronic illnesses and dental specializations, provide an intriguing but perplexing picture of the field (*Kumar et al., 2020*; *Ahirwar, Bhargava & Gupta, 2021*). Although age is a common factor

**Table 9  Pairwise comparison of random blood sugar levels with referral of the patients.**

| | | Normal patient | Referred to hospital for diabetes | Referred to hospital for hypertension | Referred to hospital for both diabetes and hypertension |
|---|---|---|---|---|---|
| Normal patient | Mean difference | – | −57.7*** | 0.166 | −60.42*** |
| | p-value | – | <.001 | 1.000 | <.001 |
| Referred to hospital for diabetes | Mean difference | | – | 57.829*** | −2.76 |
| | p-value | | – | <.001 | 0.886 |
| Referred to hospital for hypertension | Mean difference | | | – | −60.59*** |
| | p-value | | | – | <.001 |

**Notes.**
*Non significant.
**Significant.
***Highly significant.

that influences studies in field of conservative dentistry and endodontics, the frequency of these disorders presents a diversified picture (*Kumar et al., 2020*; *Ahirwar, Bhargava & Gupta, 2021*). Our research, which shows prevalences of 40.6% and 24.0% for type 2 DM and HTN, respectively, emphasizes the importance of age. These numbers, however, are quite distinct from others and may indicate differences in the patient population (*Ahirwar, Bhargava & Gupta, 2021*).

*Kumar et al. (2020)* adds another level of complexity by highlighting the possibility of a bidirectional relationship between type 2 DM and HTN, which increases the therapeutic implications of both conditions in dental settings. Even though their stated prevalences are different from ours, there is an interesting congruence shown in the age-specific rise in HTN, especially in the 41–50 age range. The compelling picture that this comparative analysis presents shows that, while illness frequency varies according to local demography and referral procedures, age serves as a unifying factor across studies. This confusing picture may ultimately become apparent if we dig deeper and take into account variables like gender and socioeconomic status in detail. This will help us identify the regional fingerprint of chronic illnesses in dental patients.

On comparison with *Tantipoj et al. (2017)* the results show different viewpoints on the relationship between dental patients and T2DM, with an emphasis on age and family history as reliable risk variables. Our research centers on the prevalence of T2DM overall and gender-specific differences in HTN, whereas *Tantipoj et al. (2017)* examine BMI, waist circumference, and oral health indicators. This emphasizes the necessity of individualized treatment plans for all dental specialties. Additionally, their emphasis on detecting undiagnosed hyperglycemia raises intriguing possibilities for routine visits as entry points for screening and early intervention, ultimately influencing oral health as well

as overall well-being. This suggests a potential role for dental offices in larger public health initiatives.

Due to possible complications, endodontic therapy for patients with T2DM and HTN necessitates cautious management (*Santhosh Kumar et al., 2020*; *Ahirwar, Bhargava & Gupta, 2021*; *Rodríguez-Rodríguez et al., 2021*). While HTN presents difficulties with intraoperative blood pressure spikes and vascular complications, uncontrolled type 2 diabetes mellitus hinders wound healing, raises the risk of infection, and impairs immunological function (*Zhang et al., 2023*). Thorough preoperative evaluations, encompassing a review of medical history and continuous monitoring of blood pressure, are essential to prevent complications (*Zhang et al., 2023*). Results can be maximized by putting stress-reduction strategies into practice, closely monitoring blood pressure, and providing careful postoperative care (*Zhang et al., 2023*). Patients with type 2 DM and HTN can get endodontic therapy that produces the best possible clinical outcomes if these issues are dealt early on.

Dental care offers an additional possibility for early systemic health intervention, extending beyond managing oral health. The intriguing significance of dentists in hypertension prevention is highlighted by their frequent interactions with patients who have undiagnosed or uncontrolled hypertension (HBP) (*López-López et al., 2011*). Dentists are first responders against hypertension, assisting physicians with coordinated treatment and preoperative blood pressure tests. Regular dental checkups also contribute to the early identification and referral of undiagnosed disorders, highlighting the vital role oral healthcare professionals play in the management of illnesses like type 2 diabetes (*Simon et al., 2023*). Patients with T2DM have considerable changes in salivary content, which raises the incidence of dental caries, according to research by *Singh et al. (2016)*.

Patients may experience discomfort or worry during the access-opening session, which can activate the sympathetic nervous system and affect blood pressure and heart rate. Rather than extrinsic causes like local anesthetics, other studies link these alterations to the body's endogenous catecholamines, such as adrenaline and noradrenaline (*Takahashi et al., 2005*; *Verberne et al., 2016*). There are differing opinions, though; some contend that the adrenaline in the local anesthetic solution could possibly be a factor in variations in heart rate and blood pressure. Although adrenaline prolongs the effects of local anesthetic, it can also have unintended side effects by constricting blood vessels and increasing blood flow, which can raise blood pressure and heart rate (*Greene, Lalonde & Seal, 2021*).

Our analysis confirms earlier findings that the prevalence of high blood pressure is 12.6% (*Lacruz et al., 2015*). It's interesting to note that, in line with previous research, among our group, which had a mean age of 44.58 years, similar participants were referred for hypertension, and a sizable number were ignorant of their illness. Among patients at screened endodontic clinics, *Bogari & Bakalka (2016)* discovered a significant prevalence of hypertension (63.7%), with more than half being unaware of their illness. According to *Saeedi et al. (2019)*, type 2 diabetes is a very frequent condition that affects 9.0% of women and 9.6% of men worldwide. Its frequency increases with age. Furthermore, *Katz & Rotstein (2021)* observed that periapical abscesses were more prevalent in people with secondary hypertension, highlighting the complex relationship between dental results and

systemic health. These findings highlight the imperative role that dental screenings play in identifying and treating undetected medical disorders and enabling integrated health care.

Ensuring the dental health and general well-being of senior patients highlights the critical role endodontic therapy plays, as it puts the demands of the patient and short-term objectives ahead of long-term stability and aesthetics. Majority of the elderly patients have positive sentiments toward root canal therapy because they have less discomfort, can preserve more of their original teeth, and feel better about themselves. Endodontists may, however, run into technical difficulties, such as restricted or calcified pulp chambers, which might result in erroneous pulp vitality tests and further issues. Endodontists are authorized to give prophylactic antibiotics when it is necessary for the patient's well-being. Endodontists can improve elderly patient comfort by optimizing appointment scheduling, offering ergonomic assistance, and utilizing magnification and transillumination methods for accurate treatment. For patients to comprehend and cooperate with treatment procedures, it is equally critical to communicate these guidelines in a kind, compassionate and clear manner (*AlRahabi, 2019*).

The public's awareness of these diseases is crucial for ensuring their health and safety worldwide. This can be achieved through various means such as organizing health camps, performing skits, and displaying posters. These measures would help educate people about these diseases and their prevention, thereby reducing the risk of outbreaks and promoting a healthier population worldwide.

The residents of Surabaya, one of East Java's densely populated cities also face the burden and challenges posed by the prevalent socio-economic and cultural factors. Inability to access proper healthcare measures due to financial constraints and uneven distribution of resources impedes early diagnosis and detection of illnesses and prevents their proper and timely interventions. Additionally, the cultural beliefs, influence of other people's thoughts and people's own ill habits and practices regarding diet, exercise and healthcare seeking behavior may also have a profound effect on their lifestyle and their overall health. Furthermore, disparities in education levels and health literacy can affect individuals' understanding of the importance of preventive measures and regular dental check-ups in managing these chronic diseases. Addressing these socioeconomic and cultural determinants is crucial for implementing effective public health interventions aimed at reducing the burden of DM and HTN and improving access to dental care in Surabaya City.

Patients with T2DM and HTN if permitted to undergo dental treatment can have serious complications which can endanger the life of the patients. Poorly controlled type 2 diabetes can compromise the ability of the tissues to heal thereby increasing the risk of infection and impaired wound healing following dental procedures. Fluctuating levels of random blood sugar levels as seen in patients with uncontrolled type 2 diabetes mellitus can lead to conditions like hypoglycemia and hyperglycemia thereby complicating the procedure.

Similarly, dental treatment administered to hypertensive patients without proper intervention precaution and management of the condition can land up in fatal consequences. Most of the hypertensive patients are on medications like aspirin which must be stopped for at least 5–7 days prior to undergoing procedures like endodontic

microsurgery, root canal treatments and dental extractions. Moreover, certain medications used to manage hypertension may interact with local anesthesia or other drugs administered during dental treatment necessitating the careful monitoring and adjustment of the treatment plans. Uncontrolled hypertension can also increase the risk of cardiovascular complications such as hypertensive crisis or stroke during dental procedures. Additionally, individuals with uncontrolled hypertension may experience anxiety or stress during dental visits which can exacerbate the blood pressure fluctuations.

Looking at the profound impact of T2DM and HTN in regards to oral health, it becomes imperative for every dental practioner to rule out these comorbidies before starting the dental treatment. Patients diagnosed with these conditions should be directed to a qualified medical practioner for appropriate management including oral anti- diabetic drug therapy with medications like sitagliptin, metformin, glimepiride before undergoing with the dental treatment.

Similarly, effective control of elevated blood pressure and referral of hypertensive patients to their consulting cardiologist is of utmost importance in providing an effective treatment. It is the duty of the consulting cardiologist to modify the dosage, duration or make other necessary changes in the schedule of anti-hypertensive drugs like aspirin and provide their patients with medical fitness so as to be deemed safe for undergoing dental treatment.

## CONCLUSIONS

Our study concludes by highlighting the high rates of hypertension and type 2 diabetes mellitus among endodontic patients in the conservative dentistry department of Surabaya. The results emphasize the need for dental professionals to incorporate systemic health factors into customized treatment programs in order to increase its practical relevance. In addition to improving treatment results, this proactive strategy establishes dental offices as key locations for the early diagnosis of type 2 diabetes and hypertension. In conservative dentistry, our study supports a move toward more complete and integrated healthcare methods by encouraging frequent screenings during dental appointments and more patient awareness.

## LIMITATIONS

Although our cross-sectional method offers useful snapshot data on the prevalence of the disease, we acknowledge that longitudinal studies could provide a more thorough understanding of the pathways that lead to the disease and the effects of treatment, especially in relation to endodontic therapy and the onset or progression of DM and HTN.

**Glossary**

| | |
|---|---|
| **T2DM** | Type 2 Diabetes Mellitus |
| **HTN** | Hypertension |
| **AP** | Apical Periodontitis |

### Funding

The authors received no funding for this work.

### Competing Interests

Ajinkya M. Pawar is an Academic Editor for PeerJ.

### Author Contributions

- Meet Manihar conceived and designed the experiments, performed the experiments, analyzed the data, prepared figures and/or tables, authored or reviewed drafts of the article, and approved the final draft.
- Dian Agustin Wahjuningrum conceived and designed the experiments, performed the experiments, prepared figures and/or tables, and approved the final draft.
- Shreya Manihar conceived and designed the experiments, prepared figures and/or tables, and approved the final draft.
- Ajinkya M. Pawar conceived and designed the experiments, performed the experiments, authored or reviewed drafts of the article, and approved the final draft.
- Jatin Atram conceived and designed the experiments, performed the experiments, prepared figures and/or tables, authored or reviewed drafts of the article, and approved the final draft.
- Kulvinder Banga conceived and designed the experiments, prepared figures and/or tables, and approved the final draft.
- Alexander Maniangat Luke conceived and designed the experiments, prepared figures and/or tables, and approved the final draft.
- Firas Elmsmari conceived and designed the experiments, prepared figures and/or tables, and approved the final draft.

### Human Ethics

The following information was supplied relating to ethical approvals (*i.e.*, approving body and any reference numbers):

This study was undertaken and executed in adherence with the principles of the Helsinki Declaration, guaranteeing the welfare, rights, and privacy of all participants. The Universitas Airlangga Faculty of Dental Medicine Health Research Ethical Clearance Commission reviewed and approved the research protocol, which received the reference number 1176/HRECC.FODM/X/2023. All participants supplied informed consent after being fully told about the study's objective, methods, potential risks, and benefits. Throughout the research procedure, participants' confidentiality and identities were scrupulously preserved.

## Ethics

The following information was supplied relating to ethical approvals (*i.e.*, approving body and any reference numbers):

The Universitas Airlangga Faculty of Dental Medicine Health Research Ethical Clearance Commission reviewed and approved the research protocol, which received the reference number 1176/HRECC.FODM/X/2023.

## Data Availability

The raw data is available in the Supplementary File.

## Supplemental Information

Supplemental information for this article can be found online at http://dx.doi.org/10.7717/peerj.17638#supplemental-information.

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
