# Peer review of "Prevalence of type 2 diabetes mellitus and hypertension in patient’s visiting the conservative dentistry and endodontics department: a cross-sectional study in Surabaya City"

_PeerJ, doi:10.7717/peerj.17638_

## Round 0.1 · original submission · Major Revisions

Dear author,

Thank you for submitting your manuscript titled "Health Trends in Patients Seeking Endodontic Treatment: A Cross-Sectional Exploration of Type 2 Diabetes Mellitus and Hypertension Prevalence in Surabaya City" to PeerJ. While we appreciate the significance of your research, after careful review by our expert reviewers, it has been determined that your manuscript requires significant revisions before it can be considered for publication.

Key areas for improvement highlighted by the reviewers include:

Refinement of the title for conciseness and alignment with study objectives.
Clear discussion on the bidirectional relationship between Type 2 DM/HTN and oral health, particularly in the context of endodontic therapy, in the introduction.
Providing a concise summary of key findings and addressing socio-economic and cultural factors influencing disease prevalence in the discussion section.
Enhancing methodology description, including addressing STROBE guidelines, clarifying patient recruitment details, and providing post hoc information for significant findings.
Rewriting the discussion section to focus on key findings, adhering to ethical guidelines, standardizing units of measurement, and providing precise definitions for technical terms.

We look forward to receiving your revised manuscript. Should you have any questions or require further assistance, please do not hesitate to contact us. Thank you for your continued interest in publishing with PeerJ.

The Editor,

Reviewer 1 ·

Basic reporting

By analyzing the prevalence of diabetes mellitus (DM) and hypertension (HTN) among patients seeking endodontic treatment, the research fills a significant gap in the literature and offers insightful information on the relationship between chronic systemic illnesses and oral health.
The study's conclusions underscore the need of taking age and other systemic health aspects into account when planning dental treatments, which has practical ramifications for dental professionals. The focus on customized treatment plans has the potential to enhance endodontic therapy patient care and clinical results.

Recommended Suggestions:
In particular, the discussion section, which discusses the contrast between the study's findings and earlier literature, could use more clarity in its presentation. A more systematic approach that concisely summarizes the study's main conclusions and how they connect to other studies would be beneficial for this part. This discussion would be simpler for readers to follow and comprehend the implications of the study's findings if it were divided into separate subsections based on theme similarity or chronological sequence.

Another suggestion, the clarity and accuracy of the study's reporting would be improved by making sure that the language used throughout the manuscript is consistent and by giving precise definitions for phrases like "referred patients" and "normal patients." Standardizing units of measurement and including a glossary for technical jargon would also help readers who are not familiar with the topic to comprehend what is being said.

Experimental design

The study implemented a cross-sectional methodology, which is ideal for looking at illness prevalence rates in a particular group at one particular period. Effective data collection and analysis are made possible by this strategy, which offers insightful information on the prevalence of hypertension (HTN) and diabetes mellitus (DM) among Surabaya patients seeking endodontic treatment.
By receiving permission from the Universitas Airlangga Faculty of Dental Medicine Health Research Ethical Clearance Commission and gaining informed consent from participants, the paper exemplifies ethical integrity. Respecting the Helsinki Declaration's tenets protects the rights, welfare, and privacy of study participants, strengthening the study's ethical foundation.

Recommended Suggestions:
Incorporating longitudinal follow-up might provide insights into illness patterns and changes over time, whereas the cross-sectional strategy just offers a snapshot of disease prevalence at a particular point in time. In order to better understand the causative pathways and treatment effects, longitudinal studies would allow the investigation of temporal connections between endodontic therapy and the onset or progression of DM and HTN. One possible drawback is that it can be brought up in the comment area.

Validity of the findings

The study's conclusions have important therapeutic implications for dentists since they highlight the need of taking age and other systemic health issues into account when treating endodontic patients. After finding that endodontic patients in Surabaya had significantly higher rates of both diabetes mellitus (DM) and hypertension (HTN), the study emphasizes the need for customized treatment plans in order to maximize patient results and guarantee patient safety.
From a methodological standpoint, the study shows strict adherence to ethical guidelines, which includes obtaining approval and adhering to the principles of the Helsinki Declaration. It also uses standardized procedures to measure blood pressure and blood sugar levels, which improves the validity and reliability of the reported prevalence rates.
The study's strong methodology and broad and varied sample, which came from Surabaya's Department of Conservative Dentistry and the surrounding communities, support the findings' generalizability and provide insightful information outside of the dental field.
These findings support multidisciplinary cooperation between dentists and medical professionals that specialize in managing chronic diseases in order to promote early identification, treatment, and management of diabetes mellitus (DM) and hypertension (HTN), ultimately leading to better patient outcomes.

Additional comments

Timely Contribution: In a city which is known for its urbanization and possibility of getting the lifestyle diseases, there is a compelling need to understand the incidence of chronic illnesses such as diabetes mellitus (DM) and hypertension (HTN) among patients seeking endodontic treatment. This study answers this requirement. By carrying out this inquiry, the study closes a significant gap in the literature and offers current information that can guide healthcare practices and policies meant to enhance patient care and results.

Interdisciplinary Collaboration: The significance of interdisciplinary collaboration in healthcare is shown by the study's focus on the relationship between dental treatment and systemic health conditions. The project fosters collaboration between dental practitioners and medical specialists to handle patients' complicated health issues fully by integrating knowledge from both professions, hence promoting a holistic approach to patient care.

Prospects for Further Research: The thorough analysis of DM and HTN prevalence among endodontic patients provided by this study establishes the foundation for further research activities. Longitudinal patterns, causal connections, and therapies targeted at enhancing outcomes for individuals with various chronic illnesses might all be investigated in further research. Furthermore, the study establishes a standard for looking into related problems in different dental specializations and geographical areas, creating opportunities for additional research and development in the field of dental public health.
General Comments:
I would like to appreciate the authors for carrying out such an excellent study on the prevalence of hypertension and diabetes mellitus among Surabaya patients seeking endodontic treatment. Your research illuminates a crucial but sometimes disregarded component of dental treatment, making a substantial contribution to the discipline. It's really impressive how extensive your inquiry was and how methodologically rigorous the study was throughout.
The promptness of your contribution and its potential to inspire future research initiatives in this field especially excite me. As a catalyst for more research and development in dental public health, your work establishes a high bar for excellence in dental research.

Reviewer 2 ·

Basic reporting

*The introduction provides a thorough overview of the connections between chronic conditions like type 2 diabetes mellitus (Type 2 DM) and hypertension (HTN) with oral health, laying a strong groundwork for the proposed research. However, there's an opportunity for refinement in terms of focus, clarity, and direct alignment with the study objectives and title.

*As for the title of the article, it could be made more concise, such as: "Prevalence of Type 2 Diabetes Mellitus and Hypertension in Surabaya City: A Cross-Sectional Study."

*I suggest the authors to focus more clearly on the bidirectional relationship between Type 2 DM/HTN and oral health, and how it specifically relates to endodontic therapy. Many points justifying the study are discussed in the discussion section and could be brought into the introduction.

*I suggest that the authors include the hypotheses at the end of the introduction.
*I would like to suggest to the authors that they review the English text and make efforts to enhance clarity. At many points, the reading is confusing and difficult to understand.

*I would recommend beginning the discussion with a brief summary of the key findings, highlighting how they meet the proposed objectives. I suggest delving deeper into the socio-economic and cultural factors in Surabaya that influence the prevalence of type 2 diabetes and hypertension, considering the impact on access to dental care and the management of chronic diseases.

*The discussion section is overly lengthy and somewhat repetitive; I recommend that the authors strive for conciseness. While it is essential to cover all relevant aspects, a careful review to condense parts of the discussion would help maintain the reader's attention on the most important points.

Experimental design

*STROBE ("Strengthening the Reporting of Observational Studies in Epidemiology") is a guideline that aim improve the quality and transparency of observational studies, facilitating critical assessment by readers and reviewers, as well as promoting better understanding and interpretation of results. Many points outlined in the STROBE guidelines were not addressed by the authors throughout the writing of the article. I strongly suggest that the authors use this tool during the revision process.

*I suggest that the authors enhance the methodology description by adding information such as setting, locations, and relevant dates, including periods of recruitment and data collection. There is no information regarding eligibility criteria, sources and methods of participant selection. How were these participants selected? For how long was this study conducted? What were the inclusion and exclusion criteria? Could patients have other associated diseases? How were demographic data (gender, location) collected (interviews, medical records)?
*Were the recruited patients specifically referred to endodontic treatment? Variables such as pulpal diagnosis and periapical condition were not collected?

*The authors mention that those patients that had blood pressure 130/90 Hg and above or glycose plasma levels above 1260 gm/d were referred to medical care. What happens to those patients? They return to finish the treatment? According to the discussion one of the major goals of the endodontists is often to be the first contact with the patient disease; in this regard, it is important to know if these patients actually sought medical attention and, if they did, whether they returned for endodontic treatment. If this information was recorded, I suggest that the authors create a flowchart, as described in the STROBE guideline (results section item 13*).

*Table 1 of an article should generally always be a table containing descriptive information about the sample. I suggest that the authors think about adding this information.

*‘’To compare Systolic Blood Pressure, Diastolic Blood Pressure, and Random Blood Sugar Level between Gender and Location, an unpaired T-test was applied’’. I was not able to find the information regarding location. I kindly ask to the authors to check this data.

*‘’To compare between Systolic Blood Pressure, Diastolic Blood Pressure, and Random Blood Sugar Level between the referral of the patients, ANOVA was applied.’’ The authors present this result in Table 6, but although the p-values are significant, there is no mention of where this difference is (post hoc information).

*"The results show a statistically significant difference between males and females in mean systolic blood pressure, with a magnitude of 3.39372 (p=0.000)." I suggest that the authors be careful with the expression "magnitude." In this study this value (3.39372) refers to a numerical difference between the mean values, while the term "magnitude" is usually reserved to express the measure of the size or strength of the observed effect (e.g., odds ratio, risk ratio).

Validity of the findings

While the study design is not groundbreaking and does not highlight any major discoveries, the topic is engaging and holds considerable interest within the endodontic community.
The results appear to be an extension of the discussion. I suggest rewriting them in a direct manner, focusing on the key findings of the article.

---

## Round 0.2 · accepted · Accept

Dear Authors,

Thank you for your diligent revisions in response to the reviewers’ comments. I have carefully assessed the revised manuscript and am pleased to inform you that all reviewer concerns have been adequately addressed. As such, I find the current version suitable for publication.

Congratulations on your work!

Reviewer 1 ·

Basic reporting

All the suggested changes have been done in terms of language and the flow of the manuscript.

Experimental design

satisfactory experimental design which offers insightful information on the prevalence of hypertension (HTN) and diabetes mellitus (DM) among Surabaya patients seeking endodontic treatment.

Validity of the findings

The findings support multidisciplinary cooperation between dentists and medical professionals that specialize in managing chronic diseases in order to promote early identification, treatment, and management of diabetes mellitus (DM) and hypertension

Additional comments

The significance of interdisciplinary collaboration in healthcare is shown by the study's focus on the relationship between dental treatment and systemic health conditions.

Reviewer 2 ·

Basic reporting

No additional comments

Experimental design

No additional comments

Validity of the findings

No additional comments

Additional comments

I can confirm that the authors have comprehensively addressed all comments and suggestions provided. The revisions made have significantly enhanced the clarity, quality, and rigor of the manuscript.

Therefore, I consider the article suitable for publication. I congratulate the authors for conducting their work and for the scientific rigor demonstrated throughout this review process.